# Association between Happiness and Economic Status among Older Adults in Two Myanmar Regions

**DOI:** 10.3390/ijerph19063216

**Published:** 2022-03-09

**Authors:** Yuri Sasaki, Yugo Shobugawa, Ikuma Nozaki, Daisuke Takagi, Yuiko Nagamine, Masafumi Funato, Yuki Chihara, Yuki Shirakura, Kay Thi Lwin, Poe Ei Zin, Thae Zarchi Bo, Tomofumi Sone, Hla Hla Win

**Affiliations:** 1Department of International Health and Collaboration, National Institute of Public Health, Wako City 351-0197, Japan; 2Graduate School of Medical and Dental Sciences, Niigata University, Niigata City 951-8510, Japan; yugo@med.niigata-u.ac.jp (Y.S.); nakayu0821@gmail.com (Y.C.); yshira@med.niigata-u.ac.jp (Y.S.); 3Bureau of International Health Cooperation, National Center for Global Health and Medicine, Tokyo 162-8655, Japan; i-nozaki@it.ncgm.go.jp; 4Department of Health and Social Behavior, Graduate School of Medicine, The University of Tokyo, Tokyo 113-0033, Japan; dtakagi-utokyo@umin.ac.jp; 5Department of Family Medicine, Tokyo Medical and Dental University, Tokyo 113-8510, Japan; yuiko.mail@gmail.com; 6Department of Global Health and Population, Harvard T.H. Chan School of Public Health, Boston, MA 02115, USA; masafumifunato2@gmail.com; 7Department of Preventive and Social Medicine, University of Medicine 1, Yangon 245, Myanmar; kaythilwin.ktl2@gmail.com (K.T.L.); poeeizin1988@gmail.com (P.E.Z.); thaezarchibo@gmail.com (T.Z.B.); prof.hlahlawin@gmail.com (H.H.W.); 8National Institute of Public Health, Wako City 351-0197, Japan; sone.t.aa@niph.go.jp

**Keywords:** happiness, objective economic status, subjective economic status, older adults, Myanmar

## Abstract

Few studies have examined whether objective or subjective economic status (ES) has a greater association with the happiness of older adults, despite concerns regarding the growing economic cost of morbidity and their functional dependence in developing countries with aging populations. Thus, this study examined whether objective/subjective ES was associated with happiness in older adults in two Myanmar regions. A multistage random sampling procedure and face-to-face interviews were conducted in the urban and rural areas of Myanmar. The happiness of 1200 participants aged >60 years was evaluated using a single happiness score ranging from 0 (very unhappy) to 10 (very happy). The wealth index, used as an objective ES, was calculated from 17 household asset items, such as radio, washing machines, and television. Subjective ES was assessed by asking “Which of the following best describes your current financial situation in light of general economic conditions?” Responses ranged from “very difficult” to “very comfortable”. Both low objective and subjective ES were negatively associated with happiness, after adjusting for confounding variables and stratification by region (urban and rural areas). Although objective and subjective ES had similar associations with happiness in urban areas, subjective ES had a stronger association in rural areas.

## 1. Introduction

There is a growing interest in the role of well-being in improving a population’s health. The European Commission argued that current measures of economic performance, such as gross domestic product, are insufficient indicators of a society’s progress and that personal well-being should also be considered [1,2]. Furthermore, it has been argued that psychological well-being should be addressed in measures of health valuation and considered in healthcare resource allocation [1,3]. 

Happiness, defined as “the overall appreciation of one’s life-as-a-whole,” is a key marker of psychological well-being [4] and is closely related to health [1]. Many studies have reported that happiness is associated with better health outcomes such as lower mortality, reduced morbidity, and functional independence in both community-dwelling and clinical populations [5,6,7,8,9,10,11]. The association between happiness and health may become more important with age, as the prevalence of chronic illnesses increases with it [1].

Several physiological mechanisms may explain these findings [12]. Happiness appears to be inversely related to perceived stress [13] and may protect against illness by improving immune responses [4]. Happy people typically enjoy better health outcomes because they demonstrate more successful adaptation; better problem-solving skills and coping strategies; more creative, imaginative, and integrative thinking; greater resilience; and a stronger ability to deal with adversity [8,14,15].

Identifying the factors of happiness is not only important for improving personal well-being, but also for gaining insight into the factors associated with health and longevity. The recognition of these factors may lead to an elevated health status, increased life expectancy, and lower medical expenses [16]. Literature suggests that happiness is associated with having high self-perceived health [15,17], health insurance coverage [18,19], social support [15], living partners [15,20,21,22] (or not living alone) [23], lower stress [21], male sex [15,24], longevity [1,12,24], absence of depression [23], and smoking history [24].

Objective economic status (ES) is also correlated with happiness. For example, people with high ES, as measured by objective indicators such as household income, are more likely to be happier than people with low ES [15,16,17,20,21,22,25,26,27]. The results of a systematic narrative review of 71 studies demonstrated that poorer well-being was associated with objective ES, which included income, wealth, and financial assets [28]. In contrast, income was strongly associated with satisfaction, but its association with happiness was weaker in 15 countries [29]. Subjective ES (i.e., perceptions of having a higher ES or better financial status) may also be important in this context [26,30,31,32]. Studies that have examined the role of subjective socioeconomic status (SES) have provided valuable comprehensive knowledge on SES–wellbeing/health connection [33]. For example, a study in five sub-Saharan countries showed that people who were moderately to completely satisfied with their financial situation were significantly more likely to report richer health and happiness than those who were completely dissatisfied [31]. Another study demonstrated that subjective SES, compared to objective SES, is a stronger predictor of psychological functioning indicators among healthy white women [34].

In Myanmar, the proportion of the population aged ≥60 years will almost double to 15% between 2010 and 2030 and is projected to reach 25% of the overall population by 2050 [35]. The implications of the increasing rate of functional dependence in Myanmar are expected to be significant, as effective medical care systems are still in their developmental stages. In addition, depression and anxiety, which are negatively associated with psychological well-being, account for 5% of disability-adjusted life years and are among the top 10 contributors to disability in Myanmar [36,37]. However, no public health organizations in Myanmar have implemented psychological health care policies, to the best of our knowledge. Consequently, psychological health services are not prioritized in primary health care, thus preventing thousands of people from accessing the services they need [36,38]. Several health organizations provide community-based healthcare services, even in remote areas, and seek to coordinate health service provision with the central healthcare system [39]. Religious organizations are also involved in service provision, and their role is gaining importance with the increasing need for collaborative action in health care [40]. However, effective medical care systems, including mental health services, remain underdeveloped. Myanmar has been under military control for several years, and national health investment has been extremely low during this period [41,42]; it resumed in 2021.

There are few published studies on the mental health of older adults in Myanmar, owing to the country’s international isolation. Although we have investigated the association between depressive symptoms and objective/subjective ES in a previous study [38], happiness and depressive symptoms are not completely contradictory concepts: happiness does not mean exhibiting non-depressive symptoms [43]. Therefore, it is crucial to examine the factors associated with happiness among older adults in Myanmar, as happiness can buffer psychological and morbidity risks and functional dependence. Furthermore, to the best of our knowledge, few studies have examined whether objective or subjective ES has a greater association with the happiness of older adults, despite concerns about the growing economic cost of morbidity and functional dependence of older adults in developing countries with ageing populations [44].

This study aimed to examine (1) the level of happiness, (2) whether objective/subjective ES is associated with happiness, and (3) differences in this association between rural and urban areas among older adults in two regions of Myanmar.

## 2. Materials and Methods

### 2.1. Study Design and Participants

This was a baseline survey of the 2018 longitudinal study on healthy and active aging in Myanmar, which examined the predictors of physical and psychological health in community-dwelling Myanmar adults aged ≥60 years. A follow-up survey is currently ongoing.

The field sites were the Yangon and Bago regions, located 91 km northeast of Yangon. Multistage random sampling was conducted in both regions. There are 45 townships in the Yangon region and 28 in the Bago region. First, six townships were randomly selected from each region via proportional population sampling based on the population of each township. Subsequently, in Yangon, 10 wards were randomly selected from each township, while in Bago, 10 village tracts were selected from each township based on the population of each township/village tract. Finally, 10 people were randomly selected from each extracted ward/village tract using the ledger lists of residents aged ≥60 years. In rural areas, multiple villages exist within a single village tract. In such cases, one village was randomly selected from the village tract.

The differences between the wards and village tracts are related to the degree of urbanization: urban areas were defined as wards, and rural areas were defined as village tracts or villages. Wards and village tracts sometimes coexist within townships. In the survey, only wards from townships in the Yangon region and only village tracts from those in the Bago region were selected to capture the characteristics of urban and rural areas from each region. The present study considered the Yangon region representative of urban areas and the Bago region representative of rural areas [38].

The sample size was calculated using a method developed by the World Health Organization [45]. We arrive at an ideal sample size of approximately 1200, with 600 sampled from urban areas and 600 from rural areas with a 100-sample margin in respective areas. The study does not adjust for a certain anticipated response rate because the authors plan to continue collecting samples up to the target number. Further details are provided in the cohort profile of the study [46].

Trained surveyors visited homes with public health nurses from each community to conduct face-to-face interviews with study participants. In Yangon, the surveyors visited 1083 older adults, 610 of whom were at home. Ten were excluded due to refusal to participate in the survey (*n* = 6), severe dementia, or being bedridden (*n* = 4). The response rate was 98.4% in Yangon. In Bago, surveyors visited 1044 older adults, of whom 694 were at home. Ninety-four patients were excluded because of severe dementia or being bedridden; thus, the response rate was 86.5% in Bago. A total of 600 people each from the Yangon (222 men and 378 women) and Bago regions (261 men and 339 women) were surveyed (Figure 1).

### 2.2. Study Tools

A structured questionnaire for face-to-face interviews was developed for this study, following the Japan Gerontological Evaluation Study (JAGES), which is a nationwide, population-based, prospective cohort study of older community-dwelling Japanese adults [47]. The linguistic translation and validation process followed the Linguistic Validation Manual for Health Outcome Assessments [48]. The scale was first translated into English. Thereafter, it was translated into the local language and back-translated into English to ensure clarity and consistency.

Research staff was hired from the Myanmar Perfect Research Company, a group that conducts epidemiological surveys. The interviewers were recruited from a company. Before the commencement of the actual survey, a two-day training course on the research protocol, administration of the questionnaire, and ethical concerns was conducted for the interviewers.

A small pilot study was conducted before the actual survey for face validity at the Urban Health Center, Dagon Township, Yangon. Participants were adults aged >60 years who visited the outpatient clinic in the center. Twenty-five respondents were recruited who provided their consent to participate in the pilot study. During the pilot study, interviewers ensured the sequence, flow, and clarity of the study. After receiving feedback from the interviewers, the questionnaire was revised accordingly.

The inclusion criteria were an age ≥60 years and residence in a selected ward or village. The exclusion criteria were being bedridden and having severe dementia. Severe dementia was defined as an Abbreviated Mental Test score ≤ 6 [49,50].

### 2.3. Dependent Variable

Previous studies have generally measured subjective well-being using a single question such as “Taking all things together, would you say that you are very happy, pretty happy, not too happy, or not happy at all?” [16,51,52,53]. It has been shown that single-item measures of subjective well-being have moderate reliability. Happiness was assessed through the following question, which was previously validated [16,54]: “How do you rate your overall happiness level on a score of 0 for very unhappy to a score of 10 for very happy?” This scale was employed because it was the present study’s sole measure of self-perceived happiness.

### 2.4. Independent Variables

The wealth index, used as an objective economic indicator, was calculated from household asset items (radio, washing machine, TV, electric rice cooker, video/DVD player, air conditioner, electric fan, bicycle, refrigerator, motorcycle, computer, car/truck, store-bought furniture, microwave oven, personal music player, mobile phone, and Internet). The principal component score was calculated based on the participants’ possession of each item and used as the wealth index, following the method described in a previous report [55]. Subjective ES was assessed by asking the following question: “Which of the following best describes your current financial situation in light of general economic conditions?” The possible responses were (1) very difficult, (2) difficult, (3) average, (4) comfortable, and (5) very comfortable. Based on their responses, participants were categorized as “difficult or very difficult” (answering 1 or 2) or “average or higher” (answering 3–5).

### 2.5. Confounding Variables

The sociodemographic characteristics that were statistically significant with happiness scores in the univariate analysis included information regarding sex (male or female), subjective health status (excellent/good or fair/poor), illness during the preceding year (no or yes), depressive symptoms (geriatric depression scale [GDS] ≥5 or <5), educational level (no school, monastic, some/all primary school, middle/high school or higher), residential area (Yangon or Bago regions), marital status (married, widow/divorced/never married), living status (alone or not), and frequency of visits to religious facilities (less than once per week, once per week, or more). In addition, variables that were significantly related to happiness in previous studies [56,57,58] with similar analyses using similar outcomes were included: sex (male or female), social support (giving and receiving emotional and instrumental help), and religion (Buddhism or other).

### 2.6. Statistical Analysis

The mean happiness scores of the sociodemographic variables categorized as above were compared using a one-way analysis of variance test. The dependent variable (happiness level) was a categorical variable for which there was a clear ordering of the category levels, therefore, ordinal logistic regression analyses were performed to identify the factors associated with happiness. Variables with a variance inflation factor of 5 (indicating multicollinearity) were excluded. The remaining variables are included in the model. Univariate and multivariate adjusted results were expressed as the odds ratio (ORs) and adjusted odds ratio (AOR) with 95% confidence interval (CI). STATA 14 (StataCorp, College Station, TX, USA) was used to perform all statistical analyses, and the statistical significance level was set at *p* < 0.05.

## 3. Results

### 3.1. Characteristics of Happy and Unhappy Respondents

Table 1 shows the happiness scores for each socio-demographic variable. For 1200 respondents, the mean happiness score was 6.58 (±2.01). Regarding ES, respondents with low objective ES had significantly lower happiness scores than those with middle/high objective ES (6.24 points versus 6.80 points, *p* < 0.001). Similarly, respondents with low subjective ES had significantly lower scores than those with average or higher subjective ES (5.62 points versus 6.83 points, *p* < 0.001). For more details, see Table 1.

### 3.2. Associations between Objective/Subjective ES and Happiness

Of the 1182 participants analyzed for multivariate analysis (Yangon: *n* = 591; Bago: *n* = 591), low objective and subjective ES were both negatively associated with happiness even after adjusting for confounding variables (AOR: 0.69, 95% CI: 0.52, 0.91; AOR: 0.46, 95% CI: 0.35, 0.62, respectively) (Table 2). After being stratified into an urban (Yangon) and a rural area (Bago), low objective and subjective ES were both still statistically or marginally associated with happiness in the urban and rural areas, even after adjusting for the confounding variables (for low objective ES in urban areas: AOR: 0.54, 95% CI: 0.32, 0.93; for low subjective ES in urban areas: AOR: 0.55, 95% CI: 0.34, 0.91; for low objective ES in rural areas: AOR: 0.73, 95% CI: 0.53, 1.01; and for low subjective ES in rural areas: AOR: 0.43, 95% CI: 0.30, 0.62). 

## 4. Discussion

To the best of our knowledge, this is the first study to investigate whether objective or subjective ES is associated with happiness using data from urban and rural areas in Myanmar. Project data from 2018 indicated that the mean happiness score was estimated to be 6.58 (±2.01) points. Overall, the model with adjusted potential confounding factors suggested that older adults with low objective ES were more likely to have lower happiness scores than those with average or higher objective ES, and the association was similar for low subjective ES.

To compare the happiness scores among other Asian populations, the mean happiness score of the participants of the JAGES was calculated [47]; the mean happiness score for adults aged ≥65 years was estimated to be 7.22 (±1.93) points in the JAGES study (*n* = 180,324). Although the mean happiness score in this study cannot be directly compared with that of the previous study due to differences in sample sizes and age ranges, the mean happiness score in Myanmar may be relatively lower than that in Japan. This may be related to Myanmar’s relatively lower SES and higher prevalence of poverty. According to the Myanmar Living Conditions Survey from 2017, which is a different survey from the 2018 longitudinal study “Healthy and Active Aging in Myanmar,” one in four people perceived themselves as poor, and another 32% were just above the poverty line, facing the risk of falling into the poverty trap in the event of adversity [59,60]. In addition, it might be related to the relatively higher prevalence of depressive symptoms in Myanmar (22.2%) compared with the median prevalence rate of depressive symptoms among adults aged ≥60 years worldwide (10.3%) [61,62].

In this study, both low objective and subjective ES were significantly or marginally associated with a lower happiness score, even after adjusting for confounding factors (Table 2). The reason why older adults with low objective/subjective ES are less likely to be happy than those with average/high objective/subjective ES in Myanmar can be speculated. A previous systematic review and meta-analysis found a seemingly consistent and statistically significant increase in the odds of cancer, angina, asthma, depression, and comorbidity prevalence when comparing low-objective ES with middle/high-objective ES [63,64]. Similarly, the risk of coronary artery disease, hypertension, diabetes, and dyslipidemia was higher when comparing low and high subjective social statuses [65]. Consistent with previous studies, in this study, low objective/subjective ES was potentially associated with physical health due to poverty, which is associated with a lack of public services such as education, health services, access to clean water, sanitation, and clean fuel. This exacerbates the vulnerability of poor people and their perception of status differentiation, which can lead to low happiness. Although we adjusted for self-rated health, health disparities due to differences in objective/subjective ES may be associated with happiness in older adults in Myanmar. Even in Japan, which is considered an egalitarian society with relatively few inequalities in health, subjective ES was negatively associated with happiness, and objective ES attenuated this association [26].

Although a previous study indicated that low subjective ES could directly increase stress or increase vulnerability to the effects of stress more than low objective ES [34], to the best of our knowledge, this is the first study to reveal that this association is more pronounced in rural areas. One of the reasons may be its association with relative deprivation (RD). Previous studies have offered the RD hypothesis as a possible explanation for the damaging implications of inequalities in health [66,67,68,69,70]. The idea behind this hypothesis is that an individual’s health or health-related behavior is determined not only by their own resources (such as income or educational attainment), but also by their relative position in reference to those resources (i.e., how much others have versus how much they have), and by the distribution of income within society [68,70]. Subjective ES might be more strongly linked to RD than objective ES, especially in rural areas, since rural areas tend to be small and strongly united [71,72], making it easier to see the economic situation of people in the same community. Even if older adults are objectively poor, they may feel happy if there is little RD in their society. In contrast, it might be harder for them to be happy if they have high neighborhood social cohesion and feel poorer than others in rural areas.

Moreover, Myanmar is classified as a country with a critical shortage of health workers, which jeopardizes access to health services, resulting in poorer health status in people in hard-to-reach places [73]. This could be particularly detrimental for older adults whose healthcare needs are high [74]. Despite ongoing changes in rural Myanmar, a considerable rural-urban gap not only in human resources (education, knowledge, and skills), but also in access to material (land, farms, and savings) and social resources (trust-based bonds) could be associated with residents’ happiness and the psychological process of coping with chronic diseases [75]. In this context, it is possible that subjective ES was more likely to be associated with happiness than objective ES among older adults living in rural areas, although possible confounding variables were adjusted for.

The strength of this study is that, to the best of our knowledge, it is the first study to investigate happiness focusing on objective/subjective ES among older adults in Myanmar. In addition, it reported the current situation of older adults after significant circumstantial changes in Myanmar. Previous studies explored the sociodemographic status of older adults in Myanmar using survey data from 2012 [76,77,78,79] and from the national census conducted in 2014 [80]. However, the changes due to Myanmar’s democratization have occurred after surveys. Since Myanmar became a stratocracy again, this survey was unique, and no other data were available for analysis.

The findings of this study should be interpreted within the context of several limitations. First, this analysis adopted a cross-sectional design. Therefore, it may include potential confounding factors or reverse causation: causal relationships between objective/subjective ES and happiness could not be determined. We plan to employ a cohort study approach to explore potential causality. Second, the happiness and subjective ES measures were based on self-reports and a single item. Therefore, although this measure has been commonly used in previous studies and has been shown to be moderately reliable [16,38,51,52,53], the results do not necessarily translate into clinical significance for psychological health. Objective and subjective multi-item and well-established measures of happiness and ES should be considered in future studies [16]. However, the mitigation of the association of low objective/subjective ES on happiness (AOR: 0.69 and 0.46, respectively) was comparable to the aggravation of the association of low self-rated health and depressive symptoms (AOR: 0.70 and 0.37, respectively; data not shown), which has exceptional predictive validity with respect to mortality and morbidity [81,82,83]. Third, it is unknown whether these findings can be generalized beyond the Yangon and Bago regions of Myanmar. People living in the Bago region may enjoy better access to urban areas and health facilities than those living in rural areas further away from the Yangon region. Therefore, it is difficult to generalize the findings to populations in areas with limited access to urban areas and health services. However, the happiness and ES of older adults can be estimated in other regions by using the level of regional urbanization. This social epidemiological survey should be extended to include surrounding regions and states throughout the country in the future [38,84]. Finally, the survey sample excluded those who were bedridden or who had severe dementia. Assuming that the happiness of these people was lower (due to their poor health) [16], happiness may have been overestimated in the analysis, and the findings may not depict the whole picture of older adults in Myanmar.

## 5. Conclusions

In conclusion, both objective and subjective ES may be associated with happiness among older adults in Myanmar. The associations remained significant or marginally significant in both urban (Yangon) and rural (Bago) areas, even after they were stratified by region. However, subjective, rather than objective, ES had a stronger association with happiness in rural areas.

## 6. Implications and Recommendations

In addition to material wealth, subjective ES may be associated with better psychological health among older adults in Myanmar. Intervention programs should use a wider range of strategies to address the economic disparities between rural and urban areas and within communities. Regional differences in objective and subjective ESs should also be considered. Future studies should examine the effectiveness of increasing happiness levels through specific interventions that consider regional differences. As population aging is a growing issue in Myanmar, it is necessary to look beyond merely ameliorating illnesses and implementing changes that enable longevity with good physical and psychological health and happiness.

## Figures and Tables

**Figure 1 ijerph-19-03216-f001:**
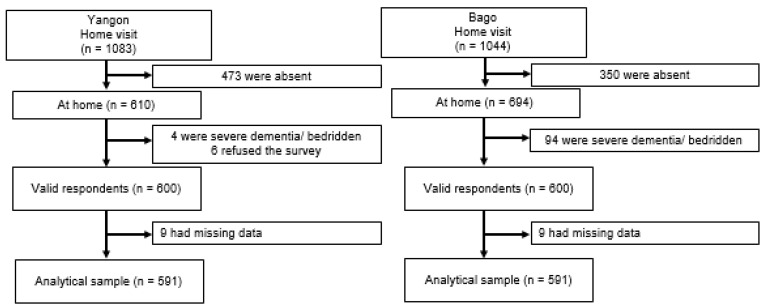
Selection of participants for a study.

**Table 1 ijerph-19-03216-t001:** Happiness scores by socio-demographic characteristics (*n* = 1200).

		N	Mean	±SD	*p*-Value
**Happiness score**		1200	6.58	2.01	
Objective SES	Middle/High	718	6.80	1.88	<0.001
(wealth index)	Low	480	6.24	2.14	
Subjective SES	Average or more	953	6.83	1.89	<0.001
(self-rated economic status)	Difficult/Very difficult	247	5.62	2.14	
Sex	Male	483	6.86	2.00	<0.001
	Female	717	6.38	2.02	
Age	60–69	670	6.61	2.00	0.25
	70–79	380	6.61	2.04	
	80+	150	6.32	2.06	
Subjective health status	Excellent/Good	354	7.10	1.89	<0.001
	Fair/Poor	846	6.36	2.01	
Illness during preceding year	No	582	6.75	2.08	<0.01
	Yes	615	6.41	1.92	
Depressive symptoms	GDS < 5	921	6.89	1.89	<0.001
	GDS ≥ 5	265	5.48	2.01	
Education	No school	104	5.72	1.94	<0.001
	Monastic	292	6.47	1.98	
	Some/Finished primary	417	6.55	2.06	
	Middle school or higher	387	6.92	1.91	
Region	Yangon	600	6.71	1.86	<0.05
	Bago	600	6.45	2.13	
Marital status	Married	642	6.77	1.93	<0.001
	Widow/Divorced/Never	558	6.35	2.07	
Living status	Alone	68	6.04	2.45	<0.05
	Not alone	1132	6.61	1.97	
Social Support					
Receiving emotional support	No	176	6.60	2.14	0.85
	Yes	1024	6.57	1.98	
Providing emotional support	No	196	6.54	2.13	0.78
	Yes	1004	6.58	1.98	
Receiving instrumental support	No	27	6.11	2.24	0.22
	Yes	1173	6.59	2.00	
Providing instrumental support	No	268	6.57	2.15	0.98
	Yes	932	6.58	1.96	
Religion	Buddhism	1147	6.57	1.99	0.47
	Other	53	6.77	2.32	
Frequency of religious visits	Less than once per week	617	6.40	1.99	<0.05
	Once per week or more	583	6.76	2.01	

*p*-value for one-way analysis of variance test; GDS = Geriatric Depression Scale; ES = Economic Status.

**Table 2 ijerph-19-03216-t002:** Univariate and multivariate adjusted association between happiness and objective/subjective ES among the older adults in two Myanmar regions.

Happiness		OR	SE	95%CI		*p*-Value		AOR	SE	95%CI		*p*-Value
Yangon & Bago	*n* = 1198						*n* = 1182					
Objective ES	Middle/High	1.00						1.00				
(wealth index)	Low	0.74	0.08	0.59	0.92	0.01		0.69	0.10	0.52	0.91	0.01
Subjective ES	Average or more	1.00						1.00				
(self-rated ES)	Difficult/Very difficult	0.33	0.48	0.25	0.44	0.00		0.46	0.07	0.35	0.62	0.00
				Pseudo = 0.0191						Pseudo = 0.0459		
*n* = 591 (Yangon)	*n* = 599						*n* = 591					
Objective ES	Middle/High	1.00						1.00				
(wealth index)	Low	0.52	0.14	0.31	0.87	0.01		0.54	0.15	0.32	0.93	0.03
Subjective ES	Average or more	1.00						1.00				
(self-rated ES)	Difficult/Very difficult	0.46	0.11	0.29	0.75	0.00		0.55	0.14	0.34	0.91	0.02
				Pseudo = 0.0100						Pseudo = 0.0359		
*n* = 591 (Bago)	*n* = 599						*n* = 591					
Objective ES	Middle/High	1.00						1.00				
(wealth index)	Low	0.76	0.12	0.55	1.04	0.08		0.73	0.12	0.53	1.01	0.06
Subjective ES	Average or more	1.00						1.00				
(self-rated ES)	Difficult/Very difficult	0.29	0.05	0.20	0.41	0.00		0.43	0.08	0.30	0.62	0.00
				Pseudo = 0.0191						Pseudo = 0.0643		

AOR: Adjusted Odds Ratio; CI: Confidence Interval; ES: Economic Status; GDS: Geriatric Depression Scale; OR: Odds Ratio; SE: Standard Error. Adjusted for age, sex, subjective health, illness during preceding year, depressive symptoms, education, region, marital status, living status, social supports, religion, frequency of religious visits.

## Data Availability

An ethics committee has placed an ethical restriction on sharing de-identified data, as the data may contain sensitive information regarding the respondents’ physical and mental health.

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
