# Peer review of "Association between Happiness and Economic Status among Older Adults in Two Myanmar Regions"

_ijerph, 2022, doi:10.3390/ijerph19063216_

Round 1
Reviewer 1 Report
The research carried out is interesting, however there are many limitations found.
- The document is difficult to read and does not contain enough information for the reader to understand the research.
The abstract should be recomposed and not include statistical data: contextualize the topic, objective, methodology and results. Do not include previously undefined acronyms.
There should be a literature review section. This lack and therefore, it is difficult to understand the concepts.
The introduction rewritten: Contextualize the topic, what has been studied? What gap is found in the literature? What is the novelty of the research? State the objective clearly.
- The main limitation is the methodology used: I consider that this is poor to achieve the proposed objective and therefore obtain relevant results and conclusions.
- There is no discussion of the results.
- There are no relevant conclusions.
My personal appreciation is that the research presented is poor and no relevant conclusions are obtained that allow enriching the literature and it does not allow for practical implications to be drawn. For all these reasons, my decision is to reject the document.
Reviewer 2 Report
The paper meets the aim and scope of the Journal, it is a good example of the interdisciplinary research. The idea to investigate links between well-being and economic status using the case of population aged 60+ is in line with emerging demographic issues. The variables are in line with research objectives, they have been justified based on the essential set of previous studies in the field. Main findings are scientifically sound. In general, the research contributes to the body of knowledge.
The main drawbacks of the research which should be corrected:
(1) in some parts of the text the sample size differs – see highlights on pages 5 (1200 respondents) and 7, Table 2 (n is 1182), the same is typical for the description of samples of two different regions (600 or 591). To be honest, such equal number of respondents in two different regions with different level of the response rate looks rather surprising; Authors should check their data and eliminate mismatches;
(2) Authors mention different years of the survey on pages 3 and 8 (2017 and 2018 respectively). Are these surveys different and devoted to other issues of happiness estimation? If yes, authors should state it clearly;
(3) using the data of 2018 in the current period of time requires additional justification. I can suppose that this survey was unique and no other data for analysis are available. In any case, Authors should explain the usage of the dataset of 2018;
(4) the details about ‘objective/subjective’ can be omitted in the title. Comparisons using different approaches to mease the economic status are provided on the main text, it’s enough;
(5) the Authors justify their position using essential Reference list, however, recent publications in the field are rare. It is recommended to update the References with appropriate improvements in the Discussion section – see notes;
(6) All over the text the requirements for the citations are unmet. See instructions for authors: https://www.mdpi.com/journal/ijerph/instructions
"In the text, reference numbers should be placed in square brackets [ ], and placed before the punctuation; for example [1], [1–3] or [1,3]. For embedded citations in the text with pagination, use both parentheses and brackets to indicate the reference number and page numbers; for example [5] (p. 10). or [6]" (pp. 101–105).
Instead, authors use inappropriate style (1, 2)…
(7) More comments see in the text.

Reviewer 3 Report
This is an interesting study on the association between happiness and SES. However, I have some concerns about the study that require the author's revision:
- First, I have some concern regarding the measure of happiness. The authors should elaborated further in details the validity of the measure. This is especially important as a single-item measure is used. More justification is necessary. It is also unclear why the authors did not use a more established scale to measure happiness.
- I don't think that the objective and subjective economic status measures were sufficiently explained. Please provide more justification on the decision to use the specific items. It is also unclear why the authors did not use a more well-established subjective social status measure such as the MacArthur Scale of Subjective Social Status
- All of the confounding variables used as covariates in the adjusted model should be justified.
- It is unclear to me why ordinal logistic regression was used instead of linear regression. It is not clear how the analysis is suitable for the nature of the current dependent variable.
- The authors used many causal terms throughout the manuscript. A major revision is necessary to rectify this.
- I would appreciate if the material and data is made open access (e.g. on the Open Science Framework) as this will facilitate meta-analysis (which the authors benefit from too)
Round 2
Reviewer 1 Report
The authors have made considerable effort. The document has improved considerably. Although I think that it still has important limitations that are difficult to solve, I also consider that the current state may be interesting for readers and, therefore, I accept the document in this form.
Reviewer 3 Report
I appreciate the authors' effort to address my previous comments. I have a few more concerns that I hope the authors can address:
- I believe that the authors should clarify how the categorize happiness in their study. The authors mentioned that happiness was assessed
through the following question, which was previously validated [16, 53]: “How do you rate your overall happiness level on a score of 0 for very unhappy to a score of 10 for very happy?” However, they claimed that the the dependent variable (happiness level) was a categorical variable. I think it is important to clarify how the happiness level was categorized into a binary variable. - It is also very important for the authors to clarify whether there is any overlap between the current manuscript with Sasaki et al. (2021). It seems to me that both papers are very similar and using the same sample. It might not be ethical to just swap depressive symptoms to happiness for publication without a good acknowledgement and clarification. It will be good for the authors to clarify and elaborate in details of this overlap and highlight the novelty of the current study.
Sasaki Y, Shobugawa Y, Nozaki I, Takagi D, Nagamine Y, Funato M, et al. Association between depressive symptoms and objective/subjective socioeconomic status among older adults of two regions in Myanmar. PLoS One. 2021;16(1):e0245489.
3. In the limitation section, it will be good for the authors to acknowledge the possibility of potential confounds and reverse causation. Reverse causation is always a possibility in depression or happiness research especially in a cross-sectional study. Here are some references:
Crossley, C. D., & Stanton, J. M. (2005). Negative affect and job search: Further examination of the reverse causation hypothesis. Journal of Vocational Behavior, 66(3), 549-560. Hartanto, A., Quek, F. Y., Tng, G. Y., & Yong, J. C. (2021). Does social media use increase depressive symptoms? A reverse causation perspective. Frontiers in Psychiatry, 12, 335. Rohrer, J. M., & Lucas, R. E. (2020). Causal effects of well-being on health: it's complicated. https://psyarxiv.com/wgbe4/download?format=pdf Rohrer, J. M. (2018). Thinking clearly about correlations and causation: Graphical causal models for observational data. Advances in methods and practices in psychological science, 1(1), 27-42.
